# Effects of Multi-Strain Probiotics on Immune Responses and Metabolic Balance in *Helicobacter pylori*-Infected Mice

**DOI:** 10.3390/nu12082476

**Published:** 2020-08-17

**Authors:** Chun-Che Lin, Wei-Chiao Huang, Chiu-Hsian Su, Wei-De Lin, Wen-Tzu Wu, Bi Yu, Yuan-Man Hsu

**Affiliations:** 1Department of Internal Medicine, China Medical University Hospital, China Medical University, Taichung 404333, Taiwan; D83949@mail.cmuh.org.tw; 2School of Medicine, China Medical University, Taichung 404333, Taiwan; 3Department of Nutrition, China Medical University, Taichung 404333, Taiwan; ariel79658@gmail.com; 4Department of Biological Science and Technology, China Medical University, Taichung 404333, Taiwan; adaga0806@hotmail.com; 5Department of Medical Research, China Medical University Hospital, China Medical University, Taichung 404333, Taiwan; weide@mail.cmu.edu.tw; 6School of Post Baccalaureate Chinese Medicine, China Medical University, Taichung 404333, Taiwan; 7Department of Health and Nutrition Biotechnology, Asia University, Taichung 413305, Taiwan; irenewu@asia.edu.tw; 8Department of Animal Science, College of Agriculture and Natural Resources, National Chung Hsing University, Taichung 402204, Taiwan; byu@dragon.nchu.edu.tw

**Keywords:** *Helicobacter pylori*, *Lactobacillus fermentum*, *Lactobacillus casei*, *Lactobacillus rhamnosus*, amino acids, fatty acids

## Abstract

Chronic inflammation caused by *Helicobacter pylori* infection increases the risk of developing gastric cancer. Even though the prevalence of *H. pylori* infection has been decreased in many regions, the development of antibiotic resistance strains has increased the difficulty of eradicating *H. pylori*. Therefore, exploring alternative approaches to combat *H. pylori* infection is required. It is well-known that probiotic therapy can improve *H. pylori* clearance. In this study, *H. pylori*-infected mice were treated with *Lactobacillus fermentum* P2 (P2), *L. casei* L21 (L21), *L. rhamnosus* JB3 (JB3), or a mixture including the aforementioned three (multi-LAB) for three days. All the lactic acid producing bacteria (LAB) treatments decreased *H. pylori* loads in the stomach and *vacA* gene expression, *H. pylori* specific immunoglobulin (Ig) A, and IgM levels in stomach homogenates, as well as serum levels of interferon-gamma and interleukin-1 beta. The multi-LAB and JB3 treatments further restored the superoxide dismutase and catalase activities suppressed by *H. pylori* infection. Furthermore, *H. pylori* infection decreased serum concentrations of 15 kinds of amino acids as well as palmitic acid. The multi-LAB treatment was able to recover the serum levels of alanine, arginine, aspartate, glycine, and tryptophan, which are all important in modulating immune functions. In addition, butyric acid, valeric acid, palmitic acid, palmitoleic acid, stearic acid, and oleic acid levels were increased. In this study, multi-LAB revealed its ability to adjust the composition of metabolites to improve health. To date, the mechanisms underlying how LAB strains crosstalk with the host are not fully understood. Identifying the mechanisms which are regulated by LABs will facilitate the development of effective therapies for infection in the future.

## 1. Introduction

It has been estimated that approximately 60.3% of the world population was infected by *Helicobacter pylori* in 2015 [1]. Despite this, a decreasing prevalence of *H. pylori* infection has been reported from different regions worldwide [2]. However, the chronic inflammation caused by *H. pylori* infection is closely related to the development of gastric cancer [3]. An effective means to eradicate *H. pylori* infection thus may reduce the risk of gastric cancer.

Standard triple therapy consisting of clarithromycin, amoxicillin, and a proton pump inhibitor (PPI) is a common anti-*H. pylori* treatment. Quadruple therapy containing two antibiotics, a PPI, and bismuth salt is frequently used as well [4]. However, the development of antibiotic resistance in *H. pylori* decreases the treatment’s efficiency [5]. Therefore, exploring alternative approaches to combat *H. pylori* infection is required. Several studies have examined the potential of probiotics to inhibit *H. pylori*. *Lactobacillus* spp. are lactic acid producing bacteria and well-characterized probiotics that have shown beneficial effects against pathogens in previous studies [6]. It has been indicated that the cultural supernatants collected from *L. acidophilus* La1 inhibited the adherence of *H. pylori* to gastric epithelial cells in vitro [7]. *L. rhamnosus* JB3 showed its ability to eliminate gastric inflammation by attenuating the virulence of *H. pylori* in mice [8]. *L. johnsonii* fermented milk reduced the risk of developing gastritis caused by *H. pylori* infection [9]. There is a growing number of studies in using probiotics to eradicate *H. pylori*. However, the underlying mechanisms are still unclear.

The host and pathogen share similar nutritional substrates and go through similar metabolic processes. The host modulates the nutrients to support an immune response to fight against the pathogen, and the pathogen depends on the nutrients for its survival. Therefore, the pathogenesis of an infection is profoundly affected by the metabolic cross-talk between host and pathogen [10]. Our previous study showed that 5 × 10^7^ colony-forming units (CFUs) of *L. rhamnosus* JB3 revealed its potential for treating *H. pylori* infection in a mice model [8]. In this study, 10^7^ CFUs of *L. rhamnosus* JB3 and two *Lactobacillus* strains were administrated to infected mice. Not only the *H. pylori* burden and immune responses, but also the alteration of amino acids and fatty acids were monitored in order to investigate the therapeutic effects and explore the possible mechanisms of *Lactobacillus* strains in combating *H. pylori* infection.

## 2. Materials and Methods

### 2.1. Bacterial Strains, Cell Lines, and Culture Conditions

The *H. pylori* reference strain 26,695 (ATCC 700392) isolated from a gastritis patient and verified by whole-genome sequencing was purchased from the American Type Culture Collection (ATCC; Manassas, VA, USA). *H. pylori* was grown on Brucella agar (Taiwan Prepared Media-TPM, Taipei, Taiwan) and supplemented with 5% sheep blood (Taiwan Prepared Media-TPM, Taipei, Taiwan) at 37 °C for 48–72 h under microaerophilic conditions. Three lactic acid bacteria (LAB) strains *Lactobacillus fermentum* P2 (P2) [11], *L. casei* L21 (L21) [12], and *L. rhamnosus* JB3 (JB3) [8] were cultured in deMan-Rogosa-Sharpe (MRS) broth (BD Biosciences, Franklin Lakes, NJ, USA) at 37 °C overnight. L21 and JB3 were then transferred to a fresh and equal volume of brain-heart infusion (BHI) medium (BD Biosciences, Franklin Lakes, NJ, USA) and incubated at 37 °C for a further 3 h, while P2 was then transferred to Nutrient Broth (BD Biosciences, Franklin Lakes, NJ, USA) for a further 1 h. Bacterial cells were collected by centrifugation at 13,000× *g* for 2 min and washed twice with sterilized phosphate buffered saline (pH 7.2), and then adjusted to the indicated colony-forming units (CFUs) for animal experiments.

### 2.2. The Mice Infection Model

A total of 60 male C57BL/6 mice weighing between 20–22 g were obtained from the National Laboratory Animal Center (Taipei, Taiwan). They were maintained in a pathogen-free environment and used for our study when they reached 6 weeks of age. The mice were housed in an air-conditioned room (25 ± 2 °C) with a relative humidity of 40–70% and were subjected to a 12-h light/dark cycle. They had ad libitum access to tap water and a standard laboratory rodent diet. All animal-based experimental protocols were approved by the Institutional Animal Care and Use Committee of China Medical University (Taichung, Taiwan; approval no. 2016–089) and performed according to the ethical rules and laws of China Medical University. The mice were randomly divided into 6 groups (10 mice per group) and fed a basal diet (Prolab RMH 2500, 5P14; LabDiet, St. Louis, MO, USA) for 1 week prior to being used in this study. To ensure *H. pylori* infection, all of the mice, apart from those in the control group, were infected with 1 × 10^9^ CFUs *H. pylori* 26,695 using gavage needles every other day for 3 doses. Then, 10^7^ CFUs of each LAB strain, P2, L21, or JB3 strain was dissolved in 0.2 mL of phosphate buffered saline (PBS) and administered orally by gavage needles every day to each mouse for three days after infection and referred to as P2, L21, and JB3 groups, respectively. The multi-LAB group received the treatment containing 1 × 10^7^ CFUs of each LAB strain (a total of 3 × 10^7^ CFUs) per mouse. The control group received the basal diet without infection and was administered only water instead of LAB strains. The infection group was infected by *H. pylori* and received the basal diet, but did not receive any treatment. Stomach and blood samples were collected from all of the groups the day after the last treatment was administered. The mice were humanely euthanized via CO_2_ asphyxiation. Blood was taken directly from their hearts via a microsyringe and centrifuged at 6000× *g* for 10 min. Serum was collected and stored at −20 °C. The stomach samples were homogenized in 1.0 mL of sterile saline, with the aid of a tissue homogenizer, at −20 °C. The homogenates were then subjected to mRNA isolation using the total RNA Miniprep Purification Kit (GMBiolab Co., Ltd., Taichung, Taiwan) described in the following section. Serum and stomach homogenates were subjected to determine the expression of interferon (IFN)-gamma, interleukin (IL)-1beta, immunoglobulin (Ig) A and IgM levels, and superoxide dismutase (SOD) and catalase activities.

### 2.3. Reverse Transcription Quantitative Polymerase Chain Reaction (RT-qPCR) Analysis

The total mRNA of stomach homogenates was isolated using the total RNA Miniprep Purification Kit (GMBiolab Co., Ltd., Taichung, Taiwan), and reverse transcription (RT) was performed using the MMLV reverse transcription kit (Protech Technology Enterprise Co., Ltd., Taipei, Taiwan). Both kits were used according to their manufacturers’ instructions. The oligonucleotide primers used for RT corresponded with the murine gene sequences and were synthesized by Mission Biotech Co., Ltd. (Taipei, Taiwan). RT-qPCR was performed at the following conditions: 10 min at 95 °C, 40 cycles of 15 s at 95 °C, and 1 min at 60 °C using 2 × Power SYBR Green PCR Master Mix (Applied Biosystems; Thermo Fisher Scientific, Inc.) and 200 nM forward and reverse primers. Primers used for amplifying 16S rRNA and *vacA* gene of *H. pylori*, and GAPDH mRNA of mice were designed in a prior study [8]. Each assay was run on an Applied Biosystems QuantStudio 3 Real-Time PCR system (Thermo Fisher Scientific, Inc. Waltham, MA, USA) and the fold-changes in expression were derived using the comparative ΔΔCq method [13]. The GAPDH mRNA of mice and 16S rRNA of *H. pylori* served as internal controls for sample loading and mRNA integrity as has been previously described [14].

### 2.4. Enzyme-Linked Immunosorbent Assay (ELISA) Evaluation of the Immune Responses of H. pylori-Infected Tissue and Cells

IFN-gamma and IL-1 beta were measured using mouse IFN-gamma and IL-1 beta Uncoated ELISA kits (Invitrogen, Inc., Waltham, MA, USA). IgA and IgM were measured using a goat anti-mouse IgA and IgM HRP-conjugated antibody (Bethyl Laboratories, Inc., Montgomery, TX, USA). SOD and catalase activity were determined by assay kits (Cayman Chemical, Inc., Ann Arbor, MI, USA). All of these kits were used following their manufacturer’s instructions. Each sample was analyzed individually. Results were calculated as the mean of triplicate readings and expressed as fold-change compared to the control group.

### 2.5. Amino Acids and Acylcarnitines Analysis

Blood amino acid and acylcarnitine profiling and quantitation was performed by electrospray tandem mass spectrometry (ESI-MS/MS) and modified as previously described [15,16]. Briefly, blood spot samples were collected from mice at the test time points and separately preserved on filter papers (Standardized Schleicher & Schull filter-paper S&S 903; Dassel, Germany). Two 1/8-inch circles were punched out of each blood spot (5.4 μL) on the filter paper and then placed in a flat bottom 96-well block. Then 100 μL of methanol extraction buffer containing standard stable-isotope labeled amino acids and acylcarnitines standard sets (Cambridge Isotope Laboratories Inc., Tewksbury, MA, USA) was added and the mixture which was then shaken for 20 min to extract amino acids and acylcarnitines in sample. Subsequently, the extract was evaporated under a gentle stream of dry nitrogen. The residue in the tube was derivatized using 100 μL of 3 N butanolic HCl (Regis Technologies, Morton Grove, IL, USA) and incubated at 65 °C for 15 min. The excess butanolic HCl was removed using a hot nitrogen gas blower; the derivatized residue was reconstituted in 100 μL of acetonitrile-water (50:50 by volume). The processed tube was covered with aluminum foil and placed in an autosampler tray for ESI-MS/MS analysis (API-2000, Sciex, Framingham, MA, USA). Data processing was performed with Analyst version 1.4.1 (Sciex, Framingham, MA, USA).

### 2.6. Statistical Analysis

The differences between the mean values of the treatment and infection groups were evaluated by Student’s *t* test and Duncan using SAS ver. 9.4 software (SAS, Inc., Cary, NC, USA). Results were then presented as the mean ± the standard error of the mean. *p* < 0.05 was considered to indicate a statistically significant difference.

## 3. Results

### 3.1. Effects of LAB Treatments on H. pylori Colonization

*H. pylori* was detected by amplifying the 16s rRNA gene. *H. pylori* loads in the stomach of infected mice were decreased after all LAB treatments as seen in Figure 1A. Furthermore, all LAB treatments also suppressed *vacA* gene expression (Figure 1B), which is involved in the colonization and the establishment of a persistent infection in the stomach [17].

### 3.2. Effects of LAB Treatments on H. pylori-Induced Immune Response

*H. pylori*-specific IgA and IgM in serum and stomach were monitored. IgA and IgM levels in stomach homogenates both saw decreases in the groups receiving LAB treatments (Figure 2). Multi-LAB and JB3 treatment also suppressed *H. pylori*-specific IgA and IgM levels in serum.

### 3.3. Effects of LAB Treatments on H. pylori-Induced Inflammation

IFN-gamma and IL-1 beta levels in serum and stomach homogenates were both increased by infection. All treatment decreased the *H. pylori* induced IFN-gamma in serum (Figure 3A) and stomach homogenates (Figure 3B), as well as IL-1 beta levels in serum (Figure 3A). The groups receiving multi-LAB or JB3 treatment also saw the suppression of *H. pylori*-induced IL-1 beta expression in stomach homogenates (Figure 3B).

### 3.4. Effects of LAB Treatments on H. pylori-Induced Oxidative stress

*H. pylori* infection induces oxidative stress in cells [18]. SOD and catalase are involved in the detoxification of these reactive oxygen species. Therefore, their enzymatic activity in serum and stomach homogenates from *H. pylori*-infected mice were examined. *H. pylori* decreased both SOD and catalase (Figure 4) activities in serum and stomachs of infected mice. However, multi-LAB and JB3 treatments were able to increase the enzymatic activities suppressed by infection.

### 3.5. Effects of LAB Treatments on H. pylori-Infection Mediated Serum Concentrations of Ammo Acids and Fatty Acids

Infectious episodes create not only a set of symptoms, but also metabolic changes in individuals [19]. In this study, after one week of infection, the serum concentrations of certain amino acids and fatty acids were observed to have changed. However, LAB treatments modulated this change. The serum levels of 15 kinds of amino acids were suppressed by *H. pylori* infection as listed in Table 1. Multi-LAB increased the levels of alanine, arginine, aspartate, glycine, and tryptophan, which were all suppressed by infection. P2 affected the concentrations of alanine, arginine, aspartate, glycine, proline and tryptophan. L21 treatment acted on the levels of alanine, arginine, glycine, and proline, while JB3 upregulated alanine, glycine, phenylalanine, and proline concentrations as well. As a whole, LAB treatments increased the concentrations of alanine, arginine, aspartate, glycine, phenylalanine, proline, and tryptophan. Serum levels of alanine and glycine were fully restored by all the LAB treatments. P2 treatment dramatically acted on increasing alanine levels, even higher than those in the control group. Interestingly, every individual LAB treatment was able to recover the level of proline, but multi-LAB treatment was not. However, glutamate, leucine/isoleucine, methionine, serine, and threonine levels were not able to be completely recovered by any of the LAB treatments. Furthermore, cysteine is the only one which was not recovered by any of the LAB treatments. Multi-LAB also revealed the ability in increasing the serum concentrations of butyric acid (C4), valeric acid (C5), palmitoleic acid (C16:1), stearic acid (C18), and oleic acid (C18:1) compared to the control group as shown in Table 2. Moreover, *H. pylori* infection decreased the serum concentration of palmitic acid (C16). The group that received the multi-LAB treatment was able to restore its level.

## 4. Discussion

The occurrence of antibiotic resistant strains of *H. pylori* decreases the efficacy of standard triple therapy over time [20]. Therefore, alternative treatments have been applied, such as probiotic supplements. It has been indicated that multi-strains of probiotics combined with triple therapy may achieve a higher eradication rate than standard therapy (74%), with 92% for a mixture of *L. acidophilus* and *B. animalis*, and 96% for a mixture of *L. helveticus* and *L. rhamnosus*. Combining the mixtures of *L. acidophilus*, *Bifidobacterium longum* and *Enterococcus faecalis* with quadruple therapies also achieved a 97% eradication rate [21]. Our findings are in agreement with previous studies. Multi-LAB treatment showed a better therapeutic effects than the individual LAB treatment did.

Nutritional status of an individual greatly affects the progress of infection [19]. Even though *H. pylori* infection did not affect the food uptake and change the body weight of mice (data not shown), infection did affect metabolic balance. Most of the amino acids suppressed by *H. pylori* infection are important in shaping the immune system of the host. Arginine influences the innate immune responses of the host [22]. Asparagine is required for T cell activation [23], and tryptophan is also important for optimal immune responses [10]. Further, arginine is the key substrate for nitric oxide (NO) biosynthesis [24] and phenylalanine also regulates NO synthesis by leucocytes [25]. Glycine is essential for the proliferation of leucocytes and also antioxidative defense [25]. Thus, lacking the substrates for NO production might also cause a decrease in catalase and SOD activities. Our results showed that multi-LAB treatment was able to adjust the amino acid concentrations to maintain homeostasis, especially those involved in against pathogen infection. Interestingly, P2 treatment increased alanine level dramatically. Alanine involves in the gluconeogenesis [25]. Thus, P2 might play a role in the generation of glucose.

On the other hand, it has been shown that amino acids are also critical for pathogens to establish infection. The availability of asparagine, arginine, and tryptophan at the site of infection would affect the competition between host and pathogens [10]. In this study, multi-LAB and P2 treatments were able to fully recover the levels of alanine, arginine, and aspartate in blood, which might supply the infected host sufficient nutrients to fight against pathogens. Surprising, multi-LAB treatment was not able to recover the serum level of proline. It has been indicated that *H. pylori* utilizes proline as respiratory substrates in the mucous layer of the stomach [26]. Since a higher number of lactic acid bacteria were administrated to the mice receiving multi-LAB treatment than other LAB treatments, which might lead to the competition of proline sources between LABs.

Fatty acids are not only important on various cellular process but also on inhibiting the growth of a broad range of bacteria, including *H. pylori* [27]. Arachidonic (C20:0), linoleic (C18:2), and oleic acid (C18:1) disrupted the cell membranes of *H. pylori* due to their antibacterial effects [28]. Palmitoleic acid (C16:0) at 0.25 mM and oleic acid at 1 mM showed bactericidal potencies against *H. pylori* as well [29]. The multi-LAB treatment was able to increase the serum concentration of palmitoleic acid and oleic acid which might be beneficial for *H. pylori* clearance.

Dietary stearic acid (C18:0) showed an effect in reducing low-density lipoprotein (LDL) cholesterol [30], thereby reducing cardiovascular and cancer risk [31]. Furthermore, increasing the oleic acid intake improved insulin sensitivity [32] and elicited beneficial effects on both obesity and type 2 diabetes mellitus. Multi-LAB treatment significantly increased serum concentrations of stearic acid and oleic acid, which demonstrated the potential of this multi-LAB mixture for preventing cardiovascular diseases and increasing insulin sensitivity. P2 treatment could further increase the level of butyric acid (C4), which is the key energy substrate of the colonocyte and modulates the immune response in the intestine, putatively suppressing colorectal cancer [33]. That might be an extra benefit for P2 uptake.

## 5. Conclusions

*H. pylori* infection not only triggers immune responses, but also metabolic alterations of an individual. Even still, *H. pylori* did not cause obvious symptoms in mice, the serum concentrations of 15 kinds of amino acids were affected, as well as palmitic acid. The multi-LAB treatment was able to adjust the composition of amino acids and also raised the serum levels of palmitoleic acid and oleic acid to fight against *H. pylori*. P2, L21, and JB3 showed their abilities to eliminate gastric inflammation by attenuating the virulence of *H. pylori* in mice. P2 might also modulate the gluconeogenesis of the host to eradicate *H. pylori* infection. In addition, multi-LAB mixture showed its potential for preventing cardiovascular diseases and increasing insulin sensitivity by increasing stearic acid and oleic acid concentrations in serum. Further, P2 uptake increased the level of butyric acid, which provides an extra benefit for putatively suppressing colorectal cancer. To date, the mechanisms underlying how LAB strains crosstalk with the host are not fully understood. Identifying the mechanisms which are regulated by LABs will facilitate the development of effective therapies for infection.

## Figures and Tables

**Figure 1 nutrients-12-02476-f001:**
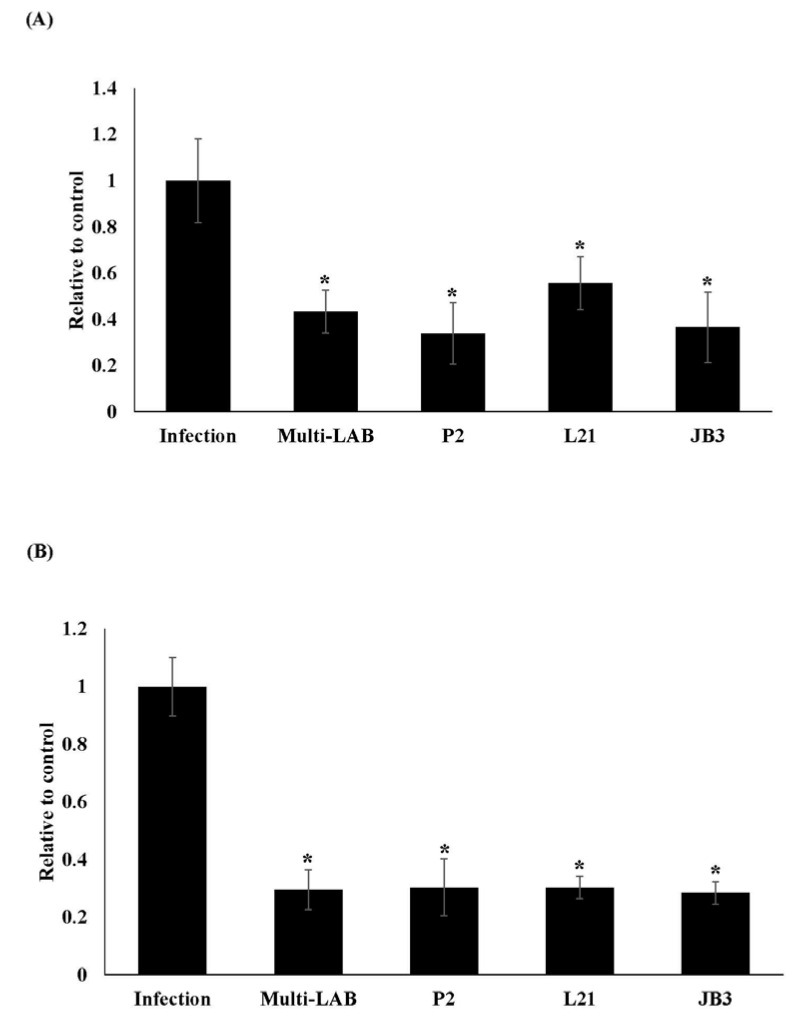
Effects of LAB strains on the expression levels of 16s rRNA (**A**) and *vacA* gene (**B**) of *H. pylori* in stomachs of mice. * Means significantly different from infection group, *p* < 0.05.

**Figure 2 nutrients-12-02476-f002:**
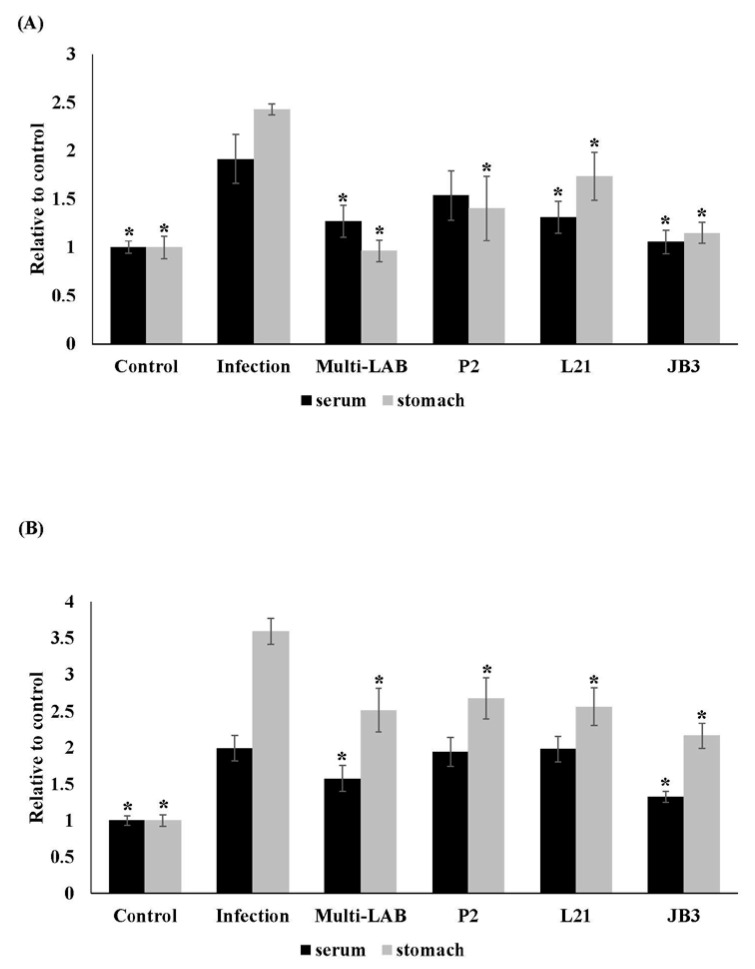
Effects of LAB strains on the *H. pylori* specific IgA (**A**) and IgM (**B**) levels in serum and stomach homogenates of mice. * Means significantly different from infection group, *p* < 0.05.

**Figure 3 nutrients-12-02476-f003:**
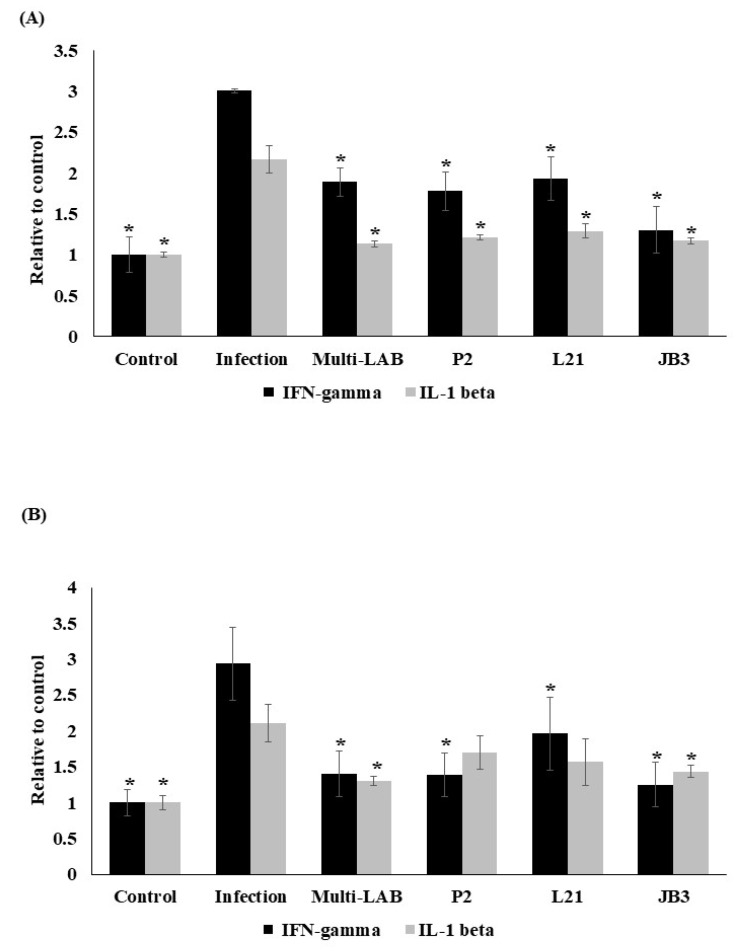
Effects of LAB strains on IFN-gamma and IL-1 beta levels in in serum (**A**) and stomach homogenates (**B**) of mice. * Means significantly different from infection group, *p* < 0.05.

**Figure 4 nutrients-12-02476-f004:**
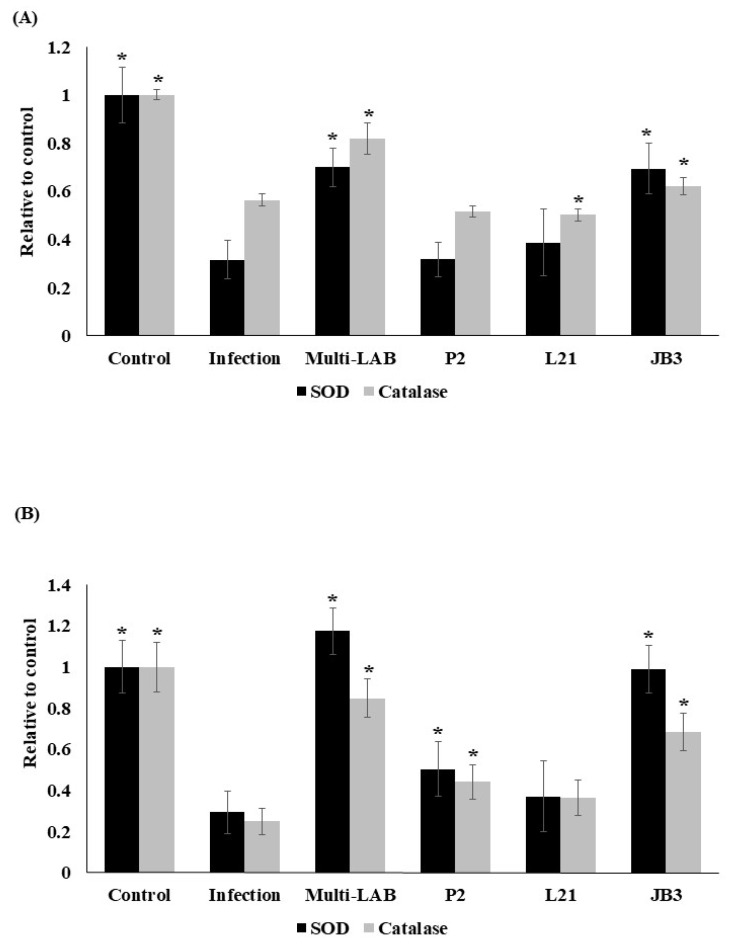
Effects of LAB strains on SOD and catalase activities in in serum (**A**) and stomach homogenates (**B**) of mice. * Means significantly different from infection group, *p* < 0.05.

**Table 1 nutrients-12-02476-t001:** Effects of LABs on amino acid concentrations (μM) in serum of *H. pylori*-infected mice. Values are mean ± SEM, *n* = 10. ^a–d^ Values in a row without a common letter differ, *p* < 0.05.

Amino Acid	Control	Infection	Multi-LAB	P2	L21	JB3
Alanine	387.27 ± 15.54 ^b^	170.69 ± 4.00 ^c^	385.90 ± 12.24 ^b^	452.97 ± 16.63 ^a^	358.01 ± 12.43 ^b^	370.54 ± 18.48 ^b^
Arginine	244.91 ± 20.99 ^a^	169.04 ± 8.65 ^c^	256.76 ± 11.04 ^a^	237.04 ± 10.01 ^a^	213.72 ± 9.35 ^a,b^	183.65 ± 15.56 ^b,c^
Aspartate	15.66 ± 2.14 ^a^	7.01 ± 0.86 ^c^	14.58 ± 0.23 ^a,b^	13.05 ± 1.21 ^a,b^	11.18 ± 0.77 ^b^	11.70 ± 0.93 ^b^
Cysteine	1.29 ± 0.16 ^a^	0.20 ± 0.14 ^b^	0.60 ± 0.26 ^b^	0.56 ± 0.27 ^b^	0.50 ± 0.24 ^b^	0.40 ± 0.12 ^b^
Glutamine	1761.05 ± 68.33 ^a^	1324.64 ± 65.06 ^c,d^	1316.64 ± 37.69 ^c,d^	1462.99 ± 42.32 ^b,c^	1565.80 ± 31.85 ^b^	1217.54 ± 67.22 ^d^
Glutamate	165.03 ± 12.18 ^a^	87.52 ± 3.80 ^c^	140.44 ± 4.06 ^b^	125.19 ± 5.97 ^b^	123.30 ± 4.47 ^b^	141.59 ± 6.07 ^b^
Glycine	310.79 ± 14.03 ^a^	260.17 ± 9.95 ^b^	327.63 ± 12.2 ^a^	309.92 ± 8.89 ^a^	324.10 ± 9.68 ^a^	317.36 ± 9.22 ^a^
Leucine/Isoleucine	268.83 ± 12.11 ^a^	183.40 ± 5.14 ^d^	191.54 ± 3.88 ^b,c^	162.92 ± 5.85 ^c^	181.73 ± 9.14 ^c^	207.68 ± 2.88 ^b^
Methionine	62.83 ± 3.87 ^a^	29.56 ± 1.18 ^c^	50.54 ± 3.07 ^b^	50.07 ± 1.37 ^b^	44.65 ± 1.36 ^b^	44.64 ± 2.20 ^b^
Phenylalanine	98.09 ± 3.98 ^a^	78.95 ± 2.88 ^c^	85.34 ± 2.17 ^b,c^	80.90 ± 2.28 ^b,c^	88.14 ± 2.78 ^b^	96.41 ± 2.43 ^a^
Proline	372.18 ± 18.59 ^a,b^	224.47 ± 10.47 ^d^	276.82 ± 10.56 ^c^	338.46 ± 11.66 ^b^	382.92 ± 20.11 ^a^	367.11 ± 11.49 ^a,b^
Serine	63.32 ± 2.44 ^a^	33.48 ± 1.69 ^d^	52.06 ± 0.87 ^b^	47.27 ± 1.55b ^c^	48.67 ± 1.19b ^c^	43.91 ± 1.77 ^c^
Threonine	83.03 ± 5.10 ^a^	42.15 ± 1.49 ^c^	56.60 ± 1.46 ^b^	58.89 ± 2.72 ^b^	59.91 ± 3.46 ^b^	57.42 ± 2.04 ^b^
Tryptophan	73.49 ± 3.20 ^a^	51.36 ± 3.64 ^c^	69.89 ± 4.23 ^a^	64.21 ± 4.30 ^a,b^	55.54 ± 2.67 ^b,c^	53.42 ± 2.63 ^c^

**Table 2 nutrients-12-02476-t002:** Effects of LABs on fatty acid concentrations (μM) in serum of *H. pylori* infected mice. Values are mean ± SEM, *n* = 10. ^a–c^ Values in a row without a common letter differ, *p* < 0.05.

Fatty Acid	Control	Infection	Multi-LAB	P2	L21	JB3
Butyric acid (C4)	0.86 ± 0.12 ^b,c^	0.81 ± 0.15 ^b,c^	1.71 ± 0.42 ^a^	1.49 ± 0.28 ^a,b^	0.64 ± 0.09 ^c^	1.06 ± 0.26 ^a,b,c^
Valeric acid (C5)	0.14 ± 0.04 ^b^	0.07 ± 0.01 ^b,c^	0.21 ± 0.02 ^a^	0.06 ± 0.03 ^c^	0.07 ± 0.02 ^b,c^	0.04 ± 0.01 ^c^
Palmitic acid (C16:0)	0.70 ± 0.08 ^a^	0.32 ± 0.04 ^b^	0.78 ± 0.10 ^a^	0.49 ± 0.02 ^b^	0.38 ± 0.02 ^b^	0.34 ± 0.06 ^b^
Palmitoleic acid (C16:1)	0.18 ± 0.03 ^a,b^	0.14 ± 0.02 ^b^	0.25 ± 0.05 ^a^	0.09 ± 0.02 ^b^	0.14 ± 0.01 ^b^	0.15 ± 0.04 ^b^
Stearic acid (C18)	0.11 ± 0.01 ^a,b^	0.09 ± 0.01 ^a,b^	0.16 ± 0.04 ^a^	0.08 ± 0.04 ^b^	0.06 ± 0.02 ^b^	0.05 ± 0.02 ^b^
Oleic acid (C18:1)	0.35 ± 0.05 ^b^	0.28 ± 0.02 ^b^	0.52 ± 0.03 ^a^	0.33 ± 0.03 ^b^	0.31 ± 0.04 ^b^	0.27 ± 0.04 ^b^

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
