# Peer review of "Effects of Multi-Strain Probiotics on Immune Responses and Metabolic Balance in *Helicobacter pylori*-Infected Mice"

_nutrients, 2020, doi:10.3390/nu12082476_

Round 1

Reviewer 1 Report

The study entitled “Effects of multi-strain probiotics on immune responses and metabolic balance in Helicobacter pylori-infected mice” reports the efficacy of lactic acid bacteria strains against H. pylori in mice in vivo. The study is an extension and confirmation of previous experiments of the authors.

Line 51: is required

Line 66-69: this is final considerations better for conclusions section. Please, explain better the aim of the study

Line 80: hr or h?

Line 81: 1 hr (space)

Line 97: to each mouse

Line 98: define better the groups

Line 135: manufacturer's instructions

Line 162: vacA italicized

Lines 165-166: what is A and what B?

Figures 3 and 4: please, combine these figures by using histograms of different color for IL-1 and γ-interferon

Line 188: their enzymatic activity

Figures 5 and 6: please, combine these figures as suggested above

Line 203: alternation?

Section 3.5: why Table 1 is not here?

Discussion: this is full of things not in the appropriate place, repetitions and reiteration of results; please, rewrite by leaving only the considerations on the significance of results. E.g.lines 222-232: this is better in the introduction, where you should also explain why you re-tested L. rhamnosus JB3. All that makes the reader to understand the rationale of your study should go in the introduction.

Discussion can be much shorter: just synthesize the possible health promoting effects of the LAB used against H. pylori.

Also the Conclusions must not be a summary of the results. Moreover: make hypotheses on which the mechanisms of action of LAB could be (also in the discussion)

Author Response

Reviewer 1

The study entitled “Effects of multi-strain probiotics on immune responses and metabolic balance in Helicobacter pylori-infected mice” reports the efficacy of lactic acid bacteria strains against H. pylori in mice in vivo. The study is an extension and confirmation of previous experiments of the authors.

Line 51: is required

Ans: Thank you for the comment. This sentence has been corrected as shown in line 54.

Line 66-69: this is final considerations better for conclusions section. Please, explain better the aim of the study

Ans: Thank you for the suggestion. The paragraph has been revised as shown in line 65-71.

Line 80: hr or h?

Ans: Thank you for the comment. Hr has been used in the whole manuscript, thus, hr was used in this sentence as well in line 82.

Line 81: 1 hr (space)

Ans: Thank you for the comment. This sentence has been corrected as shown in line 84.

Line 97: to each mouse

Ans: Thank you for the comment. This sentence has been corrected as shown in line 101.

Line 98: define better the groups

Ans: Thank you for the comment. This sentence has been revised as shown in line 99-102.

Line 135: manufacturer's instructions

Ans: Thank you for the comment. This sentence has been corrected as shown in line 136.

Line 162: vacA italicized

Ans: Thank you for the comment. This sentence has been corrected as shown in line 164.

Lines 165-166: what is A and what B?

Ans: Thank you for the comment. The figure legend has been corrected as shown in line 167.

Figures 3 and 4: please, combine these figures by using histograms of different color for IL-1 and γ-interferon

Ans: Thank you for the comment. Figure 3 has been modified.

Line 188: their enzymatic activity

Ans: Thank you for the comment. The figure legend has been corrected as shown in line 187.

Figures 5 and 6: please, combine these figures as suggested above

Ans: Thank you for the comment. Figure 4 has been modified.

Line 203: alternation?

Ans: Thank you for the comment. The figure legend has been corrected as shown in line 199.

Section 3.5: why Table 1 is not here?

Ans: Thank you for the comment. Table 1 has been included as shown in line 200.

Discussion: this is full of things not in the appropriate place, repetitions and reiteration of results; please, rewrite by leaving only the considerations on the significance of results. E.g.lines 222-232: this is better in the introduction, where you should also explain why you re-tested L. rhamnosus JB3. All that makes the reader to understand the rationale of your study should go in the introduction.

Ans: Thank you for the suggestion. Line 222-232 was moved to the introduction as shown in line 65-68.

Discussion can be much shorter: just synthesize the possible health promoting effects of the LAB used against H. pylori.

Ans: Thank you for the suggestion. the discussion section has been revised as shown in line 224-262.

Also the Conclusions must not be a summary of the results. Moreover: make hypotheses on which the mechanisms of action of LAB could be (also in the discussion)

Ans: Thank you for the suggestion. The conclusion section has been revised as shown in line 264-276.

Reviewer 2 Report

In their manuscript, Lin et al. investigate the influence of three probiotic strains on the colonization of Helicobacter pylori in mice as well as the host immune reactions after probiotic application. The work is interesting and well carried out. Before publication a few minor point should be clarified.

  1. English is rather good but a few minor mistakes should be corrected.
  2. Introduction and discussion: please point out what standard triple or quadruple therapy stands for.
  3. M&M: no background on the strain of H. pylori is given besides ATCC number. Please provide i.e. mouse pathology etc.
  4. M&M: the mice are very young when taken at 4 weeks of age. Why was this age chosen?
  5. M&M: H. pylori was applied by stomach tube. Does that mean by a gavage needle? Please clarify.
  6. M&M: how were the LAB bacteria applied? Please clarify.
  7. Results: the two tables are very difficult to understand and confusing. If possible, choose a different way to present the data or explain more extensively the way the data should be read.
  8. Discussion: why was the proline level not recovered when the multi-LAB treatment was applied? This should be discussed.

Author Response

In their manuscript, Lin et al. investigate the influence of three probiotic strains on the colonization of Helicobacter pylori in mice as well as the host immune reactions after probiotic application. The work is interesting and well carried out. Before publication a few minor point should be clarified.

English is rather good but a few minor mistakes should be corrected.

Ans: Thank you for the comment. This manuscript has been checked with a native English-speaker colleague.

Introduction and discussion: please point out what standard triple or quadruple therapy stands for. Added in introduction

Ans: Thank you for the suggestion. The compositions of triple and quadruple treatments has been added as shown in line 50-52.

M&M: no background on the strain of H. pylori is given besides ATCC number. Please provide i.e. mouse pathology etc.

Ans: Thank you for the suggestion. The background information of stain 26695 has been added in line 74-75.

M&M: the mice are very young when taken at 4 weeks of age. Why was this age chosen?

Ans: Thank you for the comment. The mice were purchased at 4 weeks of age and used when they reached 6 weeks of age. This has been corrected as shown in line 90.

M&M: H. pylori was applied by stomach tube. Does that mean by a gavage needle? Please clarify.

Ans: Yes, a gavage needle was used. The sentence has been revised as shown in line 99.

M&M: how were the LAB bacteria applied? Please clarify.

Ans: LABs were also applied by gavage needles. The sentence has been revised as shown in line 100-101.

Results: the two tables are very difficult to understand and confusing. If possible, choose a different way to present the data or explain more extensively the way the data should be read.

Ans: Thank you for the suggestion. The results of effects of LAB treatments on H. pylori-infection mediated serum concentrations of ammo acids and fatty acids has been revised as shown in line 201-216.

Discussion: why was the proline level not recovered when the multi-LAB treatment was applied? This should be discussed.

Ans: Thank you for the suggestion. this issue has been added in the discussion section as shown in line 243-247.